# Micronutrient–Antioxidant Therapy and Male Fertility Improvement During ART Cycles

**DOI:** 10.3390/nu17020324

**Published:** 2025-01-17

**Authors:** Marwa Lahimer, Severine Capelle, Elodie Lefranc, Dorian Bosquet, Nadia Kazdar, Anne Ledu, Mounir Agina, Rosalie Cabry, Moncef BenKhalifa

**Affiliations:** 1ART and Reproductive Biology Laboratory, University Hospital and School of Medicine, Picardie University Jules Verne, CHU Sud, 80000 Amiens, France; capelle.severine@chu-amiens.fr (S.C.); lefranc.elodie@chu-amiens.fr (E.L.); bosquet.dorian@chu-amiens.fr (D.B.); cabry.rosalie@chu-amiens.fr (R.C.); benkhalifa.moncef@chu-amiens.fr (M.B.); 2PERITOX-(UMR-I 01), UPJV/INERIS, UPJV, CURS, Chemin du Thil, 80025 Amiens, France; 3Eylau/Unilabs, IVF Units Cherest et la Muette, 75116 Paris, France; nadia.kazdar@unilabs.com (N.K.); anne.le.du@unilabs.com (A.L.); 4Service of Reproductive Biology, University Hospital Farhat Hached, University of Sousse, Sousse 4000, Tunisia; mounirajina09@gmail.com

**Keywords:** micronutrients, oxidative stress, male infertility, semen quality, genome decay

## Abstract

Today, accumulating evidence highlights the impact of oxidative stress (OS) on semen quality. It is considered to be a key factor contributing to the decline in male fertility. OS is detected in 30–80% of men with infertility, highlighting its strong association with impaired reproductive function and with clinical outcomes following the use of assisted reproductive technologies. Spermatozoa are particularly vulnerable to oxidative damage due to their high content of polyunsaturated fatty acids (PUFAs) and limited antioxidant defense abilities. OS arises from an imbalance between the production of reactive oxygen species and the capacity to neutralize or repair their adverse effects. Evidence indicates that OS leads to lipid peroxidation, protein oxidation, mitochondrial dysfunction, and genomic instability. Micronutrient–antioxidant therapies can play a key role in infertility improvement by neutralizing free radicals and preventing cellular damage. Many different micronutrients, including L-carnitine, L-glutathione, coenzyme Q10, selenium, and zinc, as well as vitamins complexes, are proposed to improve sperm parameters and male fertility potential. This study aims to review the impact of antioxidant supplementation on semen parameters, including sperm volume, motility, concentration, morphology, genome integrity (maturity and fragmentation), and in vitro fertilization (IVF) outcomes. Antioxidant intake and a balanced lifestyle reduce oxidative stress and mitochondrial dysfunction, enhancing the spermatogenesis and spermiogenesis processes, improving sperm quality, and protecting DNA integrity.

## 1. Introduction

Infertility is the inability to achieve pregnancy after 12 months of regular unprotected intercourse. It has a global impact on an estimated 48 million couples and around 186 million individuals [1,2], affecting around 17.5% of the adult population—roughly 1 in 6 worldwide [3]. In 1992, Carlsen et al. conducted a large meta-analysis, revealing a 50% decline in sperm count over 50 years. [4]. Numerous studies have shown similar declines in sperm parameters globally [5,6,7,8,9,10].

Understanding the causes of male infertility is essential for effective diagnosis, treatment, and prevention. The physiological role of controlled levels of reactive oxygen species (ROS) relates to their function as signaling molecules. They facilitate critical cellular processes, such as spermatogenesis, including proliferation, differentiation, and meiotic progression (Figure 1). During spermiogenesis, ROS also play a crucial role in the maturation of sperm cells. They contribute to the activation of signaling pathways that are essential for sperm maturation, capacitation (the acquisition of the ability to achieve oocyte fertilization), and the acrosome reaction (a crucial step in egg fertilization and activation) [11].

ROS involve a combination of endogenous and exogenous factors. Among the endogenous factors we find genetic causes, such as Y chromosome microdeletions and other genetic disorders [12,13,14]. Varicocele [15] and sperm infections (leucospermia) [16] are known to lead to excessive ROS generation due to cellular stress and inflammation. Conversely, elevated ROS can contribute to sperm DNA damage and impairments associated with advanced age [17,18] and hormonal imbalances [19].

ROS can be generated exogenously by environmental factors [20,21,22,23] and lifestyle factors [24]. These factors can lead to OS in the male reproductive system, arising from an imbalance between ROS production and the body’s antioxidant defenses. This imbalance adversely impacts both sperm quality and function, causing genome- and epigenome-related spermatozoa decay [25,26].

The male reproductive system, particularly sperm cells, is highly susceptible to oxidative stress due to the presence of polyunsaturated fatty acids (PUFAs) in sperm membranes, negatively affecting sperm motility and viability. Additionally, the high metabolic activity of sperm cells, which depends on oxidative phosphorylation and glycolysis for energy production, leads to the generation of ROS as byproducts. The limited antioxidant defenses in the testes and semen, such as superoxide dismutase, catalase, and glutathione, exacerbate oxidative damage, as their levels may be insufficient to effectively counteract excessive ROS production [27,28]. 

Mitochondrial DNA (mtDNA) is particularly vulnerable to ROS-induced damage due to its proximity to the electron transport chain. ROS can directly damage mtDNA, and they may also deplete mtDNA by triggering apoptosis or inhibiting replication, leading to mitochondrial dysfunction [29]. Sawyer et al. investigated the susceptibility of mtDNA to oxidative damage caused by hydrogen peroxide in two murine germ cell lines. Using a quantitative polymerase chain reaction (qPCR) assay to measure mtDNA damage, they found that mtDNA is highly sensitive to hydrogen peroxide, sustaining significant damage. These findings highlight that mtDNA is a sensitive biomarker for oxidative stress in the germ cells of mice [30]. Cells have evolved antioxidant defense systems, such as superoxide dismutase and catalase, which can neutralize ROS and prevent oxidative damage to nuclear and mtDNA [30]. 

Spermatogenesis and sperm DNA integrity have been shown to be influenced by inadequate vitamin intake [4]. Research has demonstrated that micronutrient treatments, including L-carnitine and zinc, positively impact semen parameters, sperm function, DNA integrity, epigenetic change, assisted reproductive technology (ART) outcomes, and live birth rates [31,32,33,34]. A study by Micic et al. (2019) investigated the effects of an antioxidant known as “Proxeed Plus”, containing 1000 g LC, 0.5 g ALC, 0.725 g fumarate, l g fructose, 50 mg citric acid, 10 mg zinc, 20 mg coenzyme Q10, 50 µg selenium, 90 mg vitamin C, 200 µg folic acid, and 1.5 µg vitamin B12, on sperm parameters in 175 males aged 19 to 44 years diagnosed with idiopathic oligoasthenozoospermia who failed to conceive for at least 12 months. Following a three-month treatment, the participants experienced notable improvements in sperm function and the fertilization process, including increased sperm volume, progressive motility, and vitality. Additionally, the study reported a significant reduction in DNA fragmentation index [35]. Diets rich in antioxidants and vitamins A, B, C, and E can protect sperm DNA from damage, improve sperm function, and enhance male fertility [33,36,37,38]. Furthermore, Cavallini et al. (2012) investigated the impact of medical therapy on reducing sperm aneuploidy levels and improving intracytoplasmic sperm injection (ICSI) outcomes in patients with severe idiopathic oligoasthenoteratospermia. Patients received a three-month treatment consisting of 1 g of L-carnitine and 500 mg of acetyl-L-carnitine, administered twice daily, along with a 30 mg cinnoxicam tablet every four days. The findings revealed that this antioxidant therapy significantly reduced the frequency of aneuploid sperm, leading to improved ICSI outcomes, including increased rates of biochemical and clinical pregnancies and live births [39]. 

This review focuses on the role of antioxidant supplementation in enhancing male reproductive health and potential, with a particular emphasis on improvements in semen quality, DNA integrity, and the ICSI outcomes.

## 2. Targeting Oxidative Stress in Infertility

Oxidative stress (OS) induces molecular damage in sperm, including DNA fragmentation, lipid peroxidation, and alterations in protein expression, which impair sperm function and fertilization capability [40]. Diagnostic tools, such as nitroblue tetrazolium, chemiluminescence, and oxidation–reduction potential, offer valuable insights into OS markers, including mucin-5B, 8-hydroxy-2′-deoxyguanosine, formazan crystals, and 2,2′-Azinobis. These tools also assess antioxidant levels and intracellular ROS concentrations, enabling clinicians to better diagnose OS-related male infertility. This facilitates the development of personalized treatment strategies involving antioxidants and lifestyle modifications [40]. 

The polyunsaturated fatty acids in the membrane are especially prone to attack by ROS, leading to lipid peroxidation [41]. This process damages the membrane’s integrity, affecting its fluidity, permeability, and functionality [42]. Lipid peroxidation occurs when free radicals like oxyl, peroxyl, and hydroxyl radicals steal electrons from lipids, creating reactive intermediates that trigger further reactions. This process directly damages phospholipids and can serve as a signal that triggers programmed cell death. Additionally, oxidized phospholipids play a significant role in inflammatory diseases, often driving proinflammatory changes within affected tissues [43]. 

Mitochondria are a primary source of ROS that induce apoptosis through both mitochondria-dependent and mitochondria-independent pathways [44]. Nuclear and mitochondrial ROS induce DNA damage and promote the formation of 8-hydroxy-2′-deoxyguanosine (8-OHdG), a well-recognized biomarker of oxidative stress. This molecule serves as a reliable indicator of oxidative DNA damage, providing valuable insights into cellular stress levels and the potential impact on genomic stability [45,46]. Benkhalifa et al. (2014) highlighted the critical role of mitochondria in reproductive biology, particularly in sperm and oocyte function, and explore their implications for infertility and aging. This review explains the multifaceted roles of mitochondria and their impact on sperm quality and function, addressing the significance of mitochondrial DNA (mtDNA) in oocytes [47].

Sánchez Milán et al. (2024) highlighted the impact of OS on unhealthy aging, especially in oxidative posttranslational modifications, such as the formation of trioxidized cysteine (t-Cys), where it results in permanent protein damage. Irregular t-Cys regulation disrupts critical phosphorylation signaling, causing protein alterations similar to those caused by phosphorylated serine (p-Ser) [48]. Additionally, new findings confirm that there are increased t-Cys levels in the blood proteome of aging humans, suggesting t-Cys could be a biological marker for unhealthy aging and age-related diseases [49]. 

Given the importance of Cys in maintaining redox balance, the use of Cys-rich supplements has been proposed as a potential therapeutic strategy to mitigate age-related decline. These supplements could help to restore cellular Cys levels, thereby promoting healthier aging in mammals [48]. 

Epigenetic modifications often serve as a “memory” for cellular states. ROS-induced damage can disrupt this memory by altering the epigenetic landscape. Several enzymes are involved in maintaining or modifying epigenetic marks, such as DNA methyltransferases (DNMTs), histone deacetylases (HDACs), and histone methyltransferases (HMTs). These enzymes are sensitive to oxidative stress, which alters their activity or even damages their structures [50,51]. 

Disruption to the oxidant/antioxidant balance is implicated in various human diseases, driving genetic instability, abnormal cell signaling, and alterations in gene expression that can have lasting effects on cellular function [52]. 

## 3. Micronutrients in Reproductive Medicine: A Path to Improved Fertility

Antioxidant treatment for male infertility is a growing area of interest and research, with studies investigating its potential to enhance various aspects of reproductive health [27,32,53,54,55].

Research suggests that various antioxidants play distinct roles in protecting cells and preserving fertility, highlighting the importance of investigating each component’s specific effects [56,57]. Yaris (2021) demonstrated that two antioxidant combinations, administered over six months, significantly improved sperm parameters. The first combination included L-carnitine (1 g), acetyl-L-carnitine (0.5 g), fructose (1 g), citric acid (0.5 mg), selenium (50 µg), coenzyme Q10 (20 mg), vitamin C (90 mg), zinc (10 mg), folic acid (200 µg), and vitamin B12 (1.5 µg). The second combination consisted of L-carnitine (500 mg), selenium (50 µg), coenzyme Q10 (20 mg), vitamin C (60 mg), zinc (15 mg), folic acid (400 µg), vitamin E, and ginseng (15 µg) [34]. Although these antioxidants are naturally available through a balanced diet, supplementation may be required in cases of deficiency or insufficient dietary intake (Figure 2). The combined use of multiple antioxidants can produce a synergistic effect, potentially enhancing sperm quality [53].

### 3.1. Semen Quality

Antioxidants have been investigated for their potential to improve sperm quality, including sperm concentration, motility, and morphology. Studies suggest that supplementation with antioxidants may help to reduce oxidative stress-related damage to sperm cells and enhance their overall functionality. A double-blind randomized clinical trial investigated the effect of antioxidant supplementation based on L-carnitine on conventional sperm parameters, sperm DNA fragmentation, sperm maturity, and pregnancy achievement. The study’s results showed a significant increase in sperm motility and a significant decrease in sperm fragmentation [31].

João et al. (2022) investigated the sperm static oxidation–reduction potential in the context of male infertility. They recruited 134 men with normal sperm parameters and 547 men with abnormal sperm parameters. A total of 604 patients were assessed for DNA integrity. Their findings revealed that the oxidation–reduction potential ratio was significantly different in men with abnormal sperm parameters compared to those with normal sperm parameters. Additionally, they found that the DNA integrity of men with abnormal sperm was more significantly affected by the static oxidation–reduction potential index [58]. 

L-carnitine is a naturally occurring compound and a dietary supplement that plays a crucial role in the production of energy by transporting fatty acids into cells’ mitochondria [59,60]. A study by Ma et al. (2022) found that L-carnitine enhances sperm count, morphology, and motility, in addition to increasing testosterone and LH levels [61]. A network meta-analysis of 23 randomized controlled trials, involving 1917 patients and evaluating 10 different antioxidants, revealed that L-carnitine produced the most significant improvements in sperm motility and morphology [62].

In addition, the study by Sicchieri et al. aimed to assess the effectiveness of a synthetic cryoprotectant, enhanced with L-α-phosphatidylcholine and L-acetyl-carnitine, in maintaining sperm motility and chromatin quality in cryopreserved semen samples. The study outcomes revealed an improvement in motility characteristics [63].

Coenzyme Q10 or ubiquinone plays a crucial role in the production of energy within the mitochondria, the energy powerhouse of cells. Coenzyme Q10 (a cofactor and an antioxidant) is a key component of the electron transport chain, which is responsible for generating ATP [61]. The study of Ahmed in 2022 demonstrated that Q10 improved sperm motility in patients with idiopathic OAT [64]. Similarly, the study of Cheng in 2018 reported that antioxidant treatment with L-carnitine and Q10 can improve the semen parameters and outcome of clinical pregnancy in OAT patients [62].

Zinc is the most abundant element in human semen, with concentrations much higher than seen in the bloodstream. This high zinc level in seminal plasma mainly comes from the prostate gland, reflecting its secretory activity [65]. A deficiency in zinc can lead to impaired spermatogenesis, reduced testosterone levels, and overall compromised male reproductive health [66]. It is involved in antioxidant defense, storage, production, secretion, and the function of several enzymes which play important roles in hormone regulation and meiosis during spermatogenesis [67].

A review published by Almujaydil et al. in 2023 reported that vitamins, including vitC, vitB12, Vit E, and trace elements serve as nutritional regulators, effectively decreasing oxidative stress and consequently improving sperm quality. This improvement is closely linked to the enhanced functioning of sperm mitochondria [68]. In addition, accumulating evidence shows that the administration of vitamin D in men experiencing sub-fertility has a positive effect on semen quality. This is achieved through improvements in sperm motility, enhanced sperm function, and an overall enhancement of in vitro fertility competence [69,70]. 

Moreover, selenium is an essential trace element that plays a crucial role in various physiological processes, including male fertility. It is a component of selenoproteins, which are important for maintaining sperm function and are involved in the regulation of spermatogenesis, the process of sperm production. Selenoproteins, such as glutathione peroxidases, help to protect sperm cells from oxidative stress, contributing to normal sperm development [71,72],

A network meta-analysis of randomized controlled trials (RCTs), treating a total of 23 RCTs including 10 types of antioxidants, revealed that antioxidant supplementation with L-carnitine improves sperm motility and morphology, while Omega-3 fatty acids improve sperm concentration. Additionally, coenzyme Q10 treated both sperm motility and concentration [73]. Another study, published by Yaris in 2022, involved 122 patients with idiopathic infertility, revealing that the combinations of antioxidants l-carnitine, acetyl-l-carnitine, fructose, citric acid, selenium, coenzyme Q10, vitamin C, zinc, and folic acid, administered for a period of 6 months, had a beneficial effect on sperm parameters [34].

### 3.2. Genome Integrity

Excessive damage to DNA, caused by ROS, can lead to mutations, potentially contributing to the onset of various diseases, including cancer. To counteract this, cells possess defense mechanisms to neutralize ROS and repair damaged DNA. A study by Abad et al. evaluated the impact of a 3-month oral antioxidant regimen on sperm DNA fragmentation in 20 infertile men diagnosed with asthenoteratozoospermia. The treatment included a combination of L-carnitine, vitamin C, coenzyme Q10, vitamin E, zinc, vitamin B9, selenium, and vitamin B12. The findings demonstrated a significant improvement in DNA integrity, with a marked reduction in the proportion of highly degraded sperm DNA [74]. A systematic review and meta-analysis, published in 2021, revealed a correlation between sperm DNA damage and an increased risk of miscarriage, the transmission of genetic disorders, and potential impairments in both embryonic development and subsequent postnatal outcomes [75].

Additionally, a study by Jannatifar et al., undertaken in 2022, aimed to evaluate the effects of *N*-acetyl-cysteine (NAC), alpha-lipoic acid (ALA), and their combination on sperm structure and function in asthenoteratozoospermia patients during cryopreservation. The study assessed various sperm parameters, including motility, viability, DNA fragmentation, mitochondrial membrane potential (MMP), acrosome reaction, and antioxidant enzyme activity. The authors found that the supplementation mentioned had beneficial effects on sperm DNA fragmentation in individuals with OAT (Oligo-Astheno-Teratozoospermia) [76].

Hence, the advancement of antioxidant therapy holds the potential to mitigate DNA damage caused by oxidative stress, playing a crucial role in sustaining spermatogenesis [28,77]. Another significant concern is the impact of certain types of sperm DNA damage such as aneuploidy [39], double-strand breaks (DSBs) or single-strand breaks (SSBs) [78], chromatin condensation issues [79], and telomere shortening [80]. This damage may increase the risk of miscarriage and the transmission of genetic disorders, ultimately impacting both embryonic development and post-natal outcomes [81].

A randomized clinical trial assessed the impact of antioxidant supplementation on seminal plasma antioxidant capacity, sperm DNA fragmentation, chromatin quality, and semen parameters in 48 sub-fertile men. Group 1 received antioxidant supplements, while Group 2 did not. Evaluations were performed before and after treatment using ELISA, the TUNEL assay, and aniline blue staining. The results demonstrated a significant reduction in sperm DNA fragmentation (*p* = 0.003) and histone/protamine ratio (*p* < 0.001) in the supplemented group, although no changes were found in total antioxidant capacity. These findings suggest that antioxidant therapy enhances sperm DNA integrity and chromatin structure in sub-fertile men [82]. 

### 3.3. Assisted Reproductive Technology Outcomes

In the context of IVF, the overproduction of ROS can have a direct impact on fertilization success and embryo implantation. ROS may interfere with various mechanisms necessary for successful implantation. This includes altering the endometrial receptivity, impairing the signaling pathways involved in embryo-maternal communication, and causing inflammation or immune system disturbances that impair embryo attachment [83]. These can lead to embryo development and fragmentation [84], recurrent miscarriage and spontaneous abortion [85], and implantation failure [86]. The study of Truong et al. (2017) demonstrated that the combination of Acetyl-L-Carnitine, *N*-Acetyl-L-Cysteine, and α-lipoic acid in IVF embryo cultures in mice significantly increased the blastocyst cell count. These results indicate that antioxidants can enhance embryo development when applied to IVF [87]. In this context, the study by Scaruffi et al. aimed to assess whether oral antioxidant supplementation could improve reproductive outcomes in men with low fertilization rates from previous ICSI cycles. Overall, 77 men participated, receiving a supplement combination of myo-inositol, alpha-lipoic acid, folic acid, coenzyme Q10, zinc, selenium, and B vitamins. The results demonstrated improvements in sperm concentration and motility, and a reduction in DNA fragmentation. Furthermore, fertilization rates, embryo quality, and blastocyst development were significantly enhanced following treatment. The treated group also achieved 29 clinical pregnancies, with no adverse neonatal outcomes. The findings suggest that antioxidant supplementation can improve sperm quality and increase ICSI success [88].

Research has investigated the advantages of synergizing different antioxidants to mitigate oxidative stress damage and improve the rates of successful fertilization and pregnancy while reducing the occurrence of genetic abnormalities in offspring. Furthermore, scientific evidence suggests that antioxidants possess the capacity to influence cellular signaling pathways crucial for embryo development and implantation [27,32,53,54,55]. Lahimer et al. in 2023 reported that 3-month antioxidant treatment improved the pregnancy rate and the life birth rate compared to the placebo treatment [31]. Several reviews also determined the positive impact of antioxidant therapy on achieving clinical pregnancy, whether through spontaneous conception or assisted reproduction methods [89,90]. 

Smits et al. (2019) conducted a comprehensive analysis of data from 11 studies and found an increase in clinical pregnancy rates linked to various antioxidant treatments. The reported outcomes are consistent with the finding presented in the Cochrane review, indicating a positive association between antioxidant interventions and increased rates of clinical pregnancy [91].

Mitochondria and lysosomes are interconnected, and their decline is a hallmark of aging. Research reveals that the lysosome-like vacuole supports mitochondrial health by dividing amino acids [92]. Excess cysteine levels disrupt mitochondrial respiration by limiting iron availability via an oxidant-based mechanism. However, the depletion of cysteine or supplementation with iron restore mitochondrial health. A study published in 2020 reported that cysteine toxicity is a key factor in mitochondrial aging and highlighted vacuolar compartmentation as a protective strategy to prevent amino acid-induced damage [93].

## 4. What Is Known

Research on the potential effects of antioxidants on male infertility has yielded mixed results. The efficacy of antioxidant supplementation in enhancing fertility outcomes appears to vary across studies, influenced by factors such as participant characteristics, study design, antioxidant types, and dosage. Further studies are necessary to determine the optimal dosage and duration and the specific antioxidants required for improving fertility [94]. Every person possesses a distinctive genetic composition, makes unique lifestyle choices, experiences distinct environmental exposures, and maintains an individual health status. These factors collectively impact how individuals respond to antioxidant supplementation. Differences in susceptibility to oxidative stress and baseline antioxidant levels can significantly impact the effectiveness of supplementation. Consequently, personalized approaches that consider these individual variations are likely essential to optimize the benefits of antioxidant interventions [56].

Antioxidants are known to protect cells from oxidative stress, which is linked to pathological conditions like prostate cancer. While inflammation often precedes prostate cancer development, studies suggest antioxidants may help to prevent this cancer. However, results from large-scale clinical trials have been inconsistent, complicating the assessment of their true role. Despite these mixed findings, some antioxidants show potential for prostate cancer chemoprevention, supported by promising preclinical and clinical data [95].

A study by Nateghian et al. evaluated the protective effects of Pentoxifylline (PT) and L-carnitine (LC) on sperm quality. Using 26 samples from normozoospermic men, processed via the swim-up technique, the samples were divided into three groups: an untreated control, an LC-treated group, and a PT-treated group. They were then incubated for up to 12 days at 4–6 °C. The findings revealed that PT supplementation significantly increased the percentage of motile spermatozoa compared to both the control and LC-treated groups. In contrast, LC supplementation resulted in a higher percentage of viable spermatozoa compared to the PT-treated and control groups. Additionally, the percentage of spermatozoa with normal protamine content remained stable across all groups throughout the 12-day storage period [96]. 

Prostasomes are multilayered lipid vesicles originating from prostate epithelial cells. They play a crucial role in facilitating intercellular communication between prostatic epithelial cells and spermatozoa. Prostasomes are rich in bioactive molecules and contribute to regulating key sperm functions, including motility and immunomodulation, thereby influencing fertilization capabilities [97]. Dietary modulation can influence the composition and function of seminal prostasomes, particularly abnormal seminal prostasomes. Pascoal et al., 2022 reported that consuming antioxidant bioactive food compounds (BFCs), including micronutrients and non-nutrients such as α-tocopherol, ascorbic acid, trace elements, polyunsaturated fatty acids, carnitines, coenzyme Q10, and *N*-acetylcysteine, improves spermatogenesis and fertility parameters [98]. The paternal intake of BFCs can also benefit the epigenetic signature of the offspring and restore metabolic health [98]. Imbalanced nutrition, including excessive calorie intake and a lack of fibers, vitamins, and BFCs, combined with insufficient physical activity, leads to body fat accumulation and increases the risk of diseases like cardiovascular disease [99], diabetes, and cancer [100]. Additionally, lifestyle factors such as smoking, alcohol abuse, and poor nutrition can impair sperm quality, including semen volume, motility, and overall quality, contributing to reduced male fertility [101]. 

A study conducted on 50 infertile men experiencing oxidative stress evaluated the effects of a 3-month oral antioxidant treatment. The findings revealed no significant improvements in conventional sperm parameters, including concentration, motility, and morphology [102]. Additionally, 48 infertile couples underwent Fertimax2 antioxidant treatment for a minimum of two months. The results showed no significant differences in sperm parameters between the treatment group and the control group [103,104]. 

Addressing oxidative stress in infertility presents several challenges that require careful consideration. The variability in oxidative stress profiles and individual responses to treatments highlights the need for personalized approaches, tailored to each patient [105]. Additionally, the long-term efficacy and safety of antioxidant therapies remain uncertain, necessitating further research to evaluate their impact on reproductive outcomes over time. Integrating traditional methods with innovative treatments, such as plant-based antioxidants, probiotics, and adaptogens, offers a promising avenue for holistic care. Lastly, ensuring global access to these advanced therapies is crucial, particularly in resource-limited settings, in order to promote equitable reproductive healthcare [106].

## 5. Conclusions

In this review, we show that low levels of ROS are essential during spermatogenesis and at each stage of fertilization. Oxidative stress plays a significant role in infertility, affecting both men and women. Understanding the mechanisms of infertility and developing therapeutic strategies to combat them offer promising solutions. By combining antioxidant therapies and lifestyle interventions, infertility can be alleviated. Antioxidant therapy contributes to the improvement of male fertility by reducing oxidative stress, enhancing semen parameters including sperm motility and morphology, protecting DNA integrity by reducing genome decay, increasing DNA compaction, and supporting reproductive health. It is important to mention that individual responses to antioxidant supplementation may vary, and that the underlying causes of male infertility can be diverse. Fertility specialists could incorporate micronutrient supplementation as part of personalized treatment plans to improve sperm quality and enhance the success rate of ART (assisted reproductive technology) cycles.

## 6. Limitations

While some supplements are beneficial, others may not provide the desired results for everyone. Supplements may contain contaminants, such as heavy metals, pesticides, or other harmful substances. Due to inadequate oversight during manufacturing, the purity of these products can vary. Ongoing research and innovation remain essential for effectively integrating these approaches into routine clinical practice.

## Figures and Tables

**Figure 1 nutrients-17-00324-f001:**
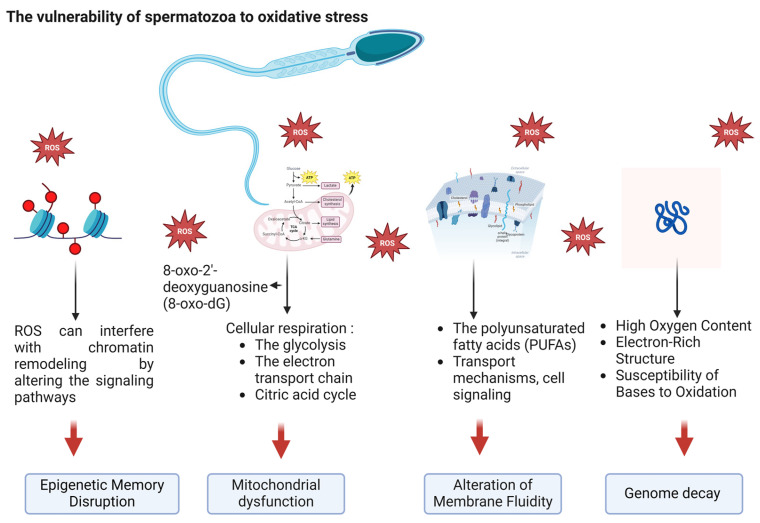
The ROS: genesis and targets in spermatozoa.

**Figure 2 nutrients-17-00324-f002:**
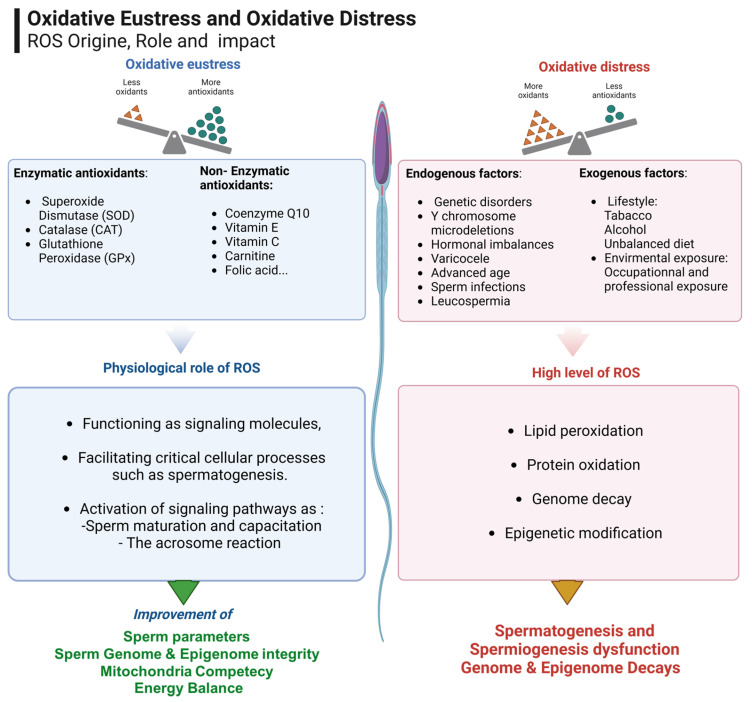
The sources of ROS, their role in spermatozoa function according to the oxidant/antioxidant imbalance, and the contribution of antioxidants to the male reproductive system.

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
