# Peer review of "Micronutrient–Antioxidant Therapy and Male Fertility Improvement During ART Cycles"

_nutrients, 2025, doi:10.3390/nu17020324_

Round 1
Reviewer 1 Report
Comments and Suggestions for Authors
The manuscript by Lahimer et al. reviews the impact of micronutrient-antioxidant therapies on various aspects of male infertility. The topic is timely and relevant; however, as it is a narrative review, some selection bias in the references included is to be expected. This important limitation should be explicitly mentioned in the manuscript. The following revisions are suggested to enhance the clarity and overall quality of the paper:
1. Title: Please consider simplifying the title. In its current form, it is overly complex and may deter potential readers.
2. Structure: Adding a list of abbreviations and a summary would greatly assist readers in navigating the review more effectively.
3. Abstract: The abstract requires revision to simplify complex sentences and improve clarity. Focus on highlighting key findings, avoid unnecessary jargon, and define essential terms for better accessibility.
4. Content Expansion: Consider incorporating new mechanisms linked to exacerbated oxidative stress that influence key signaling processes and may be targeted by the dietary therapies discussed. The following recent reference may be helpful: PMID: 39058307.
5. Prostasome Function: Discuss the effects of dietary modulation on abnormal seminal prostasomes, which are critical in male infertility. Key points to address include:
- Diets rich in antioxidants and bioactive food compounds (BFCs) can mitigate oxidative damage in the male reproductive system, potentially enhancing prostasome function (Pascoal et al., 2022).
- Nutrients such as omega-3 fatty acids, vitamins, and trace elements have been shown to improve semen quality and may positively influence prostasome efficacy (Serkis et al., 2024; Pascoal et al., 2022).
- Reducing the intake of harmful substances like alcohol and processed foods can further support reproductive health and prostasome function (Serkis et al., 2024).
6. Limitations: Highlight the limitations of nutritional supplementation in terms of purity and quality, as these are not regulated like pharmaceutical drugs. Additionally, recommend that individuals seeking to address male infertility through dietary modulation consult professional nutritional advice.
Author Response
Note: the line numbers quoted in the replies refer to the “clean” version of the revised manuscript.
We thank the reviewers for their positive feedback and hope that the revisions prompted by their comments have improved the manuscript.
Reviewer #1 comments on Manuscript ID: nutrients-3409796
Comments: The manuscript by Lahimer et al. reviews the impact of micronutrient-antioxidant therapies on various aspects of male infertility. The topic is timely and relevant; however, as it is a narrative review, some selection bias in the references included is to be expected. This important limitation should be explicitly mentioned in the manuscript. The following revisions are suggested to enhance the clarity and overall quality of the paper:
Answer: We thank the reviewer for this very positive feedback and believe that the revisions prompted by his/her comments and those of the other reviewers have improved the manuscript.
- Title: Please consider simplifying the title. In its current form, it is overly complex and may deter potential readers.
Answer: We thank the reviewer and agree that the title can be simplified and based on this comment,
- The title was changed to “Micronutrients-Antioxidants therapy and male fertility improvement during ART cycles ”
- Structure: Adding a list of abbreviations and a summary would greatly assist readers in navigating the review more effectively.
Answer: we thank the reviewer for pointing that out.
- A list of abbreviations and a graphical abstract was added to simplify the understanding of the review.
- Abstract: The abstract requires revision to simplify complex sentences and improve clarity. Focus on highlighting key findings, avoid unnecessary jargon, and define essential terms for better accessibility.
Answer: We thank the reviewer for pointing that out.
- We rewrote the abstract
- Content Expansion: Consider incorporating new mechanisms linked to exacerbated oxidative stress that influence key signaling processes and may be targeted by the dietary therapies discussed. The following recent reference may be helpful: PMID: 39058307.
Answer: We thank the reviewer for this important suggestion
- A paragraph was added, and the proposed study was cited (Sánchez Milán JA et al., 2024) Aging (Albany NY) and (Sánchez Milán JA et al., 2024) Aging Cell: Line 133- Line143 and Line 313 - 319
- Prostasome Function: Discuss the effects of dietary modulation on abnormal seminal prostasomes, which are critical in male infertility. Key points to address include:
Answer: We thank the reviewer for this suggestion and the proposed citations. We believe that it enriches our review.
- Diets rich in antioxidants and bioactive food compounds (BFCs) can mitigate oxidative damage in the male reproductive system, potentially enhancing prostasome function (Pascoal et al., 2022).
- Nutrients such as omega-3 fatty acids, vitamins, and trace elements have been shown to improve semen quality and may positively influence prostasome efficacy (Serkis et al., 2024; Pascoal et al., 2022).
- Reducing the intake of harmful substances like alcohol and processed foods can further support reproductive health and prostasome function (Serkis et al., 2024).
- A paragraph was added, citing the study by Pascoal et al. (2022) in lines 349–365.
- While we attempted to locate the citation for Serkis et al. (2024), we were unfortunately unable to find it. However, to enhance the content of this paragraph, six additional citations were included.
- Another paragraph was added Line 333 - 338
- Limitations: Highlight the limitations of nutritional supplementation in terms of purity and quality, as these are not regulated like pharmaceutical drugs. Additionally, recommend that individuals seeking to address male infertility through dietary modulation consult professional nutritional advice.
Answer: We thank the reviewer for pointing that out and we agree that the limitations must be added.
- The limitation section was added Line 398- 403
Reviewer 2 Report
Comments and Suggestions for Authors
my comments:
1. line 37: "Male infertility is the inability to achieve a pregnancy after 12 months of regular unprotected intercourse." - this is the definition of infertility, but not male infertility. The whole sentence is very poorly constructed. One can conclude that it is the man who gets pregnant. - Please correct this.
2. line 38/39: "It has a global impact on an estimated 48 million couples and around 186 million individuals, affecting around 17-20% of couples worldwide [1–3]" - this type of data should be supported by the latest publications possible. These are not very new (especially 1 and 2)
3. line 41/42: what do the authors mean by "declines in sperm parameter”: - their number in a unit of semen volume or the quality (morphology, motility, vitality, etc.) of sperm - please specify
4. line 42: "Numerous studies have shown similar declines in sperm parameter globally [5,6]" - citing only 2 studies (5 and 6) you cannot write that they are "numerous" - please make it more realistic or increase the number of references
5. line 44" what do the authors mean by "controlled reactive oxygen species" and how do they differ from "uncontrolled" - if such exist at all - please explain.
6. lines 43-50: please illustrate this with a figure (for example Figure 1 would be OK here)
7. lines 51-54: "ROS involves a combination of endogenous and exogenous factors. Among the endogenous factors are genetic causes, such as Y chromosome microdeletions and other genetic disorders [8–10], hormonal imbalances [11], varicocele [12], advanced age [13,14], sperm infections and leucospermia [15]." - This entire fragment is incomprehensible - it is not known whether all these pathologies cause larger amounts of ROS to be created in the body, or whether ROS cause all these pathologies. In both cases, I have doubts that there is a direct connection between ROS and some of the pathologies mentioned here. Please write it in a way that is understandable and substantively correct.
8. line 80:"Numerous studies" - the authors refer only to 4 studies (27-30) - these are not numerous studies - please verify this
9. lines 83/84: "A study by Saya et al. (2019) involved 175 males aged 19 to 44 years with idiopathic oligoasthenozoospermia who had failed to conceive for 12 months." - please specify, it would be more accurate to use the term "fertilization" here
10. line 85: "Proxeed Plus" - please describe exactly what this product is (composition, manufacturer, dose)
11. Figure 1 - is not indicated anywhere in the text - in my opinion it should be placed earlier - in the place where I suggested using the figure
12. Figure 1: - please indicate the source of this figure
13. line 104; line 106 - "OS" is an abbreviation - explain earlier
14. line 104-107: "Diagnostic tools reviewed provide valuable insights into OS markers, antioxidant levels, and intracellular ROS concentrations, enabling clinicians to better diagnose OS-related male infertility and personalize treatment strategies which include the use of antioxidants, and lifestyle modifications" - it would be interesting if the authors listed these "OS markers" and "diagnostic tools" - but only those that are actually validated for diagnostics and not markers and tools only for scientific research. Without providing this detailed information, this entire sentence is irrelevant to the topic of the review.
15. line 108/109: "The polyunsaturated fatty acids in the membrane are especially prone to attack by ROS, leading to lipid peroxidation." - indicate the source of this information
16. “2. Targeting Oxidative Stress in Infertility” - this entire subsection is basically a repetition of what has been written before, only in different words. Just don't repeat the same information in several places in the manuscript. Furthermore, none of the paragraphs develop the details of these issues, which should be expected from a review. The information provided is general. Nothing is explained. This is not a popular science paper, but a scientific one (for professionals, I think), and in fact it is more like a text for those interested but not familiar with the subject.
17. "3. Micronutrients in Reproductive Medicine: A Path to Improved Fertility":
a. line 138/139: "Antioxidant treatment for male infertility is a growing area of interest and research, with studies investigating its potential to enhance various aspects of reproductive health" - are these two studies the only ones that address this topic? Are there no newer reports (2023/2024)
b. Figure 2: it is not indicated anywhere in the text, it is not known what it refers to, provide the source of the figure or the information contained in it
c. lines 141-153: "Increasing evidence suggests that various antioxidants play distinct roles in protecting cells and preserving fertility, highlighting the importance of investigating each component's specific effects. ..." - the authors write that "Increasing evidence suggests [...]" and they refer to one study - this is not enough to talk about "Increasing evidence" – correct
d. lines 149-153: provide the source of this information
e. the rest of the comments for this subchapter (3) are the same as for subchapter 2 (see point 16 of the review) - too little information, too little explanation of this information, too generally written, too little literary data analyzed and everything was done too chaotically.
18. Discussion in a review is not generally necessary. Discussion serves to confront the results of one's own research with the results of research by other authors in the same field. Therefore, it is not used in a review. The review is to have an educational value. The authors did not discover anything new here, did not apply new research, did not obtain their own results. There is nothing to discuss here. What has been called a discussion here is rather a general summary. Too general, as is the entirety of this review. Please expand and reword it.
19. the conclusions are very poorly related to the presented review
20. References:
a. - more studies can be found on this topic
b. - formatting is not in accordance with the MDPI requirement
21. other comments:
a. - there is no information about how data was selected for the review.
b. - the authors use phrases such as "numerous studies", "increasing evidence", "many authors", "numerous authors" etc. in many places, at the same time they usually support such phrases with 1-3 publications - these are not numerous studies, numerous evidence, numerous authors etc. Please correct this and make it more realistic
c. - the authors did not point out the weaknesses and limitations of this review
d. - the authors did not point out what practical applications the information resulting from this review may have
The language requires refinement to be scientifically correct. Popular science phrases and mental shortcuts were used, which can be misleading. In this type of work, precise expression is required, without mental shortcuts and colloquialisms. The text uses abbreviations that have not been explained.
Author Response
Note: the line numbers quoted in the replies refer to the “clean” version of the revised manuscript.
We thank the reviewers for their positive feedback and hope that the revisions prompted by their comments have improved the manuscript.
Reviewer #2 comments on Manuscript ID: nutrients-3409796
Comments:
- line 37: "Male infertility is the inability to achieve a pregnancy after 12 months of regular unprotected intercourse." - this is the definition of infertility, but not male infertility. The whole sentence is very poorly constructed. One can conclude that it is the man who gets pregnant. - Please correct this.
Answer: We thank the reviewer for pointed this out and believe that the revisions prompted by his/her comments and those of the other reviewers have improved the manuscript.
- We agree with the reviewer comment, and we corrected the sentence Line 36
- line 38/39: "It has a global impact on an estimated 48 million couples and around 186 million individuals, affecting around 17-20% of couples worldwide [1–3]" - this type of data should be supported by the latest publications possible. These are not very new (especially 1 and 2)
Answer: We thank the reviewer for pointed this out.
- We agree re searched and found new values. The entire sentence was rectified, and new citations were added Line 38-39
- line 41/42: what do the authors mean by "declines in sperm parameter”: - their number in a unit of semen volume or the quality (morphology, motility, vitality, etc.) of sperm - please specify
Answer: we thank the reviewer for noticing such details.
- In this sentence, we refer to "declines in sperm parameters" as encompassing both aspects: the quantity of sperm in a unit of semen volume and the quality of sperm, including morphology, motility, and vitality.
- line 42: "Numerous studies have shown similar declines in sperm parameter globally [5,6]" - citing only 2 studies (5 and 6) you cannot write that they are "numerous" - please make it more realistic or increase the number of references
Answer: we thank the reviewer for pointed this out.
- Additional citations were added.
- line 44" what do the authors mean by "controlled reactive oxygen species" and how do they differ from "uncontrolled" - if such exist at all - please explain.
Answer: we thank the reviewer for this question.
- Physiological levels of ROS or controlled levels reactive oxygen species are required to activate transcription factors involved in the intracellular signaling that mediates essential physiological processes like spermatogenesis, sperm maturation, capacitation, hyperactivation, chemotaxis, acrosome reaction, and sperm–oocyte fusion. But the high ROS production can result in oxidative damage to proteins, lipids, and nucleic acids
- The sentence was rectified and the term “ levels “ was added line 43
- lines 43-50: please illustrate this with a figure (for example Figure 1 would be OK here)
Answer: we thank the reviewer for pointed this out.
- The citation of fig1 was added in line 46
- lines 51-54: "ROS involves a combination of endogenous and exogenous factors. Among the endogenous factors are genetic causes, such as Y chromosome microdeletions and other genetic disorders [8–10], hormonal imbalances [11], varicocele [12], advanced age [13,14], sperm infections and leucospermia [15]." - This entire fragment is incomprehensible - it is not known whether all these pathologies cause larger amounts of ROS to be created in the body, or whether ROS cause all these pathologies. In both cases, I have doubts that there is a direct connection between ROS and some of the pathologies mentioned here. Please write it in a way that is understandable and substantively correct.
Answer: we thank you for your feedback on this paragraph
- To clarify, the statement refers to endogenous factors that are either associated with increased production of ROS or are exacerbated by elevated ROS levels. For example, conditions like varicocele and sperm infections are known to lead to excessive ROS generation due to cellular stress and inflammation. Conversely, oxidative stress caused by elevated ROS can contribute to sperm DNA damage and impairments associated with advanced age and hormonal imbalances. However, not all pathologies listed have a direct causal link to ROS. We will rephrase the section to ensure it is clearer and more precise: Line 52- 55.
- line 80:"Numerous studies" - the authors refer only to 4 studies (27-30) - these are not numerous studies - please verify this
Answer: we thank the reviewer for pointed this out.
- The sentence was rectified Line 81
- lines 83/84: "A study by Saya et al. (2019) involved 175 males aged 19 to 44 years with idiopathic oligoasthenozoospermia who had failed to conceive for 12 months." - please specify, it would be more accurate to use the term "fertilization" here
Answer: we thank the reviewer for the comment.
- We re wrote the paragraph: Line 84-91
- line 85: "Proxeed Plus" - please describe exactly what this product is (composition, manufacturer, dose)
Answer: we thank the reviewer for pointed this out.
- We added the detail of this antioxidant treatment Line 85-87
- Figure 1 - is not indicated anywhere in the text - in my opinion it should be placed earlier - in the place where I suggested using the figure
Answer: we thank the reviewer for this detail
- we added the figure in the place where you suggested;
- Figure 1: - please indicate the source of this figure
Answer: we thank the reviewer for this detail
- We used Biorender for the illustration of the figure 1 and 2. So the figures are our production using biorender.
- line 104; line 106 - "OS" is an abbreviation - explain earlier
Answer: we thank the reviewer for this detail
- The sentence was rectified line 106
- line 104-107: "Diagnostic tools reviewed provide valuable insights into OS markers, antioxidant levels, and intracellular ROS concentrations, enabling clinicians to better diagnose OS-related male infertility and personalize treatment strategies which include the use of antioxidants, and lifestyle modifications" - it would be interesting if the authors listed these "OS markers" and "diagnostic tools" - but only those that are actually validated for diagnostics and not markers and tools only for scientific research. Without providing this detailed information, this entire sentence is irrelevant to the topic of the review.
Answer: We thank the reviewer for these helpful comments, which we believe will enhance the quality of the manuscript.
- The paragraph was rectified : line 108- 113
- line 108/109: "The polyunsaturated fatty acids in the membrane are especially prone to attack by ROS, leading to lipid peroxidation." - indicate the source of this information
Answer: we thank the reviewer for pointed his out
- The reference was added : line 116
- “2. Targeting Oxidative Stress in Infertility” - this entire subsection is basically a repetition of what has been written before, only in different words. Just don't repeat the same information in several places in the manuscript. Furthermore, none of the paragraphs develop the details of these issues, which should be expected from a review. The information provided is general. Nothing is explained. This is not a popular science paper, but a scientific one (for professionals, I think), and in fact it is more like a text for those interested but not familiar with the subject.
Answer: we thank the reviewer for pointed his out
- The section was revised and paragraphs were added Line 133- 143
- "3. Micronutrients in Reproductive Medicine: A Path to Improved Fertility":
- line 138/139: "Antioxidant treatment for male infertility is a growing area of interest and research, with studies investigating its potential to enhance various aspects of reproductive health" - are these two studies the only ones that address this topic? Are there no newer reports (2023/2024)
Answer: we thank the reviewer for pointed his out
- New references were included : line 158
- Figure 2: it is not indicated anywhere in the text, it is not known what it refers to, provide the source of the figure or the information contained in it
Answer: we thank the reviewer for pointed his out
- The figure 2 was cited in line 169
- We used Biorender for the illustration of the figure 1 and 2. So the figures are our production using biorender.
- lines 141-153: "Increasing evidence suggests that various antioxidants play distinct roles in protecting cells and preserving fertility, highlighting the importance of investigating each component's specific effects. ..." - the authors write that "Increasing evidence suggests [...]" and they refer to one study - this is not enough to talk about "Increasing evidence" – correct
Answer: we thank the reviewer for pointed his out
- The sentence was corrected in line 159
- lines 149-153: provide the source of this information
Answer: we thank the reviewer for pointed his out
- References were included, Line 161
- the rest of the comments for this subchapter (3) are the same as for subchapter 2 (see point 16 of the review) - too little information, too little explanation of this information, too generally written, too little literary data analyzed, and everything was done too chaotically.
Answer: we thank the reviewer for this comment
- We acknowledge that the information presented in this subchapter may not be as extensive as desired. We understand your concerns regarding the lack of detailed explanation and analysis. However, we worked to improve the clarity and structure of the subchapter by including a more thorough analysis of the available data and ensuring the content is more coherent and informative
- paragraph was added line 284-297
- paragraph was added line 313-319
- Discussion in a review is not generally necessary. Discussion serves to confront the results of one's own research with the results of research by other authors in the same field. Therefore, it is not used in a review. The review is to have an educational value. The authors did not discover anything new here, did not apply new research, did not obtain their own results. There is nothing to discuss here. What has been called a discussion here is rather a general summary. Too general, as is the entirety of this review. Please expand and reword it.
Answer: we thank the reviewer for pointed his out and we agree with your comment
- The section title was rectified Line 320
- Paragraphs were added Line 333- 338 and line 349- 365
- the conclusions are very poorly related to the presented review
Answer: we thank the reviewer for pointed his out
- The conclusion was rewritten.
- References:
- - more studies can be found on this topic
Answer: we thank the reviewer for pointed his out and several studies was added
- - formatting is not in accordance with the MDPI requirement
Answer: we thank the reviewer for this comment and we will revise the references according to the MDPI requirement
- other comments:
Answer: we thank the reviewer for discussing these details
- - there is no information about how data was selected for the review.
- Our article is not a systematic or meta-analysis study; therefore, we did not include details on how the data was selected for the review. However, to clarify, we used Google Scholar and PubMed NCBI for sourcing the relevant literature.
- - the authors use phrases such as "numerous studies", "increasing evidence", "many authors", "numerous authors" etc. in many places, at the same time they usually support such phrases with 1-3 publications - these are not numerous studies, numerous evidence, numerous authors etc. Please correct this and make it more realistic
- These phrases were rectified in the manuscript
- - the authors did not point out the weaknesses and limitations of this review
Answer: We thank the reviewer for pointing that out and we agree that the limitations must be added.
- The limitation section was added Line 398- 304
- - the authors did not point out what practical applications the information resulting from this review may have
- The paragraph was added Line 394- 397
Comments on the Quality of English Language
The language requires refinement to be scientifically correct. Popular science phrases and mental shortcuts were used, which can be misleading. In this type of work, precise expression is required, without mental shortcuts and colloquialisms. The text uses abbreviations that have not been explained.
Answer: we thank the reviewer for pointing that out.
- The manuscript was revised, and we agree that the abbreviations must be defined. A table of abbreviation list was added
Round 2
Reviewer 1 Report
Comments and Suggestions for Authors
The authors have succesfully addressed all my cooments and I recommend publication of the manuscript.
Reviewer 2 Report
Comments and Suggestions for Authors
I would like to thank the authors for the changes introduced in the manuscript. However, I still have some minor reservations - the list of references is not formatted according to MDPI guidelines - please correct it. I have no other comments.